# Is There a Difference in Clinical Features, Microbiological Epidemiology and Effective Empiric Antimicrobial Therapy Comparing Healthcare-Associated and Community-Acquired Vertebral Osteomyelitis?

**DOI:** 10.3390/antibiotics10111410

**Published:** 2021-11-18

**Authors:** Siegmund Lang, Astrid Frömming, Nike Walter, Viola Freigang, Carsten Neumann, Markus Loibl, Martin Ehrenschwender, Volker Alt, Markus Rupp

**Affiliations:** 1Department of Trauma Surgery, University Hospital Regensburg, Franz-Josef-Strauss-Allee 11, 93053 Regensburg, Germany; astrid.froemming@t-online.de (A.F.); Nike.Walter@klinik.uni-regensburg.de (N.W.); viola.freigang@klinik.uni-regensburg.de (V.F.); carsten.neumann@ukr.de (C.N.); markus.loibl@kws.ch (M.L.); volker.alt@ukr.de (V.A.); Markus.Rupp@klinik.uni-regensburg.de (M.R.); 2Department of Spine Surgery, Schulthess Clinic, Lenghalde 2, 8008 Zurich, Switzerland; 3Institute of Clinical Microbiology and Hygiene, University Hospital Regensburg, Franz-Josef-Strauss-Allee 11, 93053 Regensburg, Germany; Martin.Ehrenschwender@barmherzige-regensburg.de

**Keywords:** vertebral osteomyelitis, healthcare-associated infections, antimicrobial resistance, epidemiology, spine, coagulase-negative staphylococci, systemic antibiotic therapy, treatment

## Abstract

Background: Empiric antibiotic therapy for suspected vertebral osteomyelitis (VO) should be initiated immediately in severely ill patients, and might be necessary for culture-negative VO. The current study aimed to identify differences between community-acquired (CA) and healthcare-associated (HA) VO in terms of clinical presentation, causative pathogens, and antibiotic susceptibility. Methods: Cases of adult patients with VO treated at a German university orthopaedic trauma center between 2000 and 2020 were retrospectively reviewed. Patient history was used to distinguish between CA and HA VO. Susceptibility of antibiotic regimens was assessed based on antibiograms of the isolated pathogens. Results: A total of 155 patients (with a male to female ratio of 1.3; and a mean age of 66.1 ± 12.4 years) with VO were identified. In 74 (47.7%) patients, infections were deemed healthcare-associated. The most frequently identified pathogens were *Staphylococcus aureus* (HAVO: 51.2%; CAVO: 46.8%), and Coagulase-negative Staphylococci (CoNS, HAVO: 31.7%; CAVO: 21.3%). Antibiograms of 45 patients (HAVO: *n* = 22; CAVO: *n* = 23) were evaluated. Significantly more methicillin-resistant isolates, mainly CoNS, were found in the HAVO cohort (27.3%). The highest rate of resistance was found for cefazolin (HAVO: 45.5%; CAVO: 26.1%). Significantly higher rates of resistances were seen in the HAVO cohort for mono-therapies with meropenem (36.4%), piperacillin–tazobactam (31.8%), ceftriaxone (27.3%), and co-amoxiclav (31.8%). The broadest antimicrobial coverage was achieved with either a combination of piperacillin–tazobactam + vancomycin (CAVO: 100.0%; HAVO: 90.9%) or meropenem + vancomycin (CAVO: 100.0%; HAVO: 95.5%). Conclusion: Healthcare association is common in VO. The susceptibility pattern of underlying pathogens differs from CAVO. When choosing an empiric antibiotic, combination therapy must be considered.

## 1. Introduction

Hematogenous pyogenic vertebral osteomyelitis (VO) is one of the most frequent manifestation of hematogenous osteomyelitis in adults with an estimated incidence of 2.4 cases per 100,000 people in European countries [1,2]. An increasing incidence of VO has been reported, which is mainly due to two reasons. First, susceptible populations with a history of an increasing number of previous spine surgeries. Second, the improved accuracy in diagnosis results in earlier and more frequent detection of VO [3]. With increasing age, and thus related higher prevalence of chronic diseases of the European population, the incidence of vertebral osteomyelitis is expected to further increase [4,5]. The most common causative pathogen of VO in Europe is *Staphylococcus aureus* (*S. aureus*). In more than 50% of the culture-positive cases, *S. aureus* is responsible for VO, followed by up to 11–25% of VO caused by Gram-negative pathogens [6]. Insidious and indolent courses of VO often result in delayed diagnosis, which potentially leads to high morbidity and mortality [7]. Predisposing risk factors for VO are previous spine surgery, a distant infectious focus, diabetes mellitus, advanced age, intravenous drug use, HIV infection, immunosuppression, oncologic history, renal failure, rheumatological diseases, and liver cirrhosis [8]. Recently, healthcare-associated vertebral osteomyelitis (HAVO) has been reported to make a relevant share of all VO cases. HAVO reportedly lead to higher rates of mortality and higher recurrence of infection compared to community-acquired vertebral osteomyelitis (CAVO) [9]. Further, HAVO has been associated with prolonged antibiotic therapy and a decreased functional status [10,11]. One main source for HAVO is (intravascular) catheter-related and device-related bloodstream infection. Up to 34.0% of HAVO coagulase-negative staphylococci could be isolated from blood cultures [12].

In both CAVO and HAVO, empiric antibiotic therapy is often necessary. Indications range from severe or progressive neurologic deficits, hemodynamic instability, and culture-negative VO [6,13]. To achieve best possible empirical antibiotic coverage, susceptibility of the most likely pathogens being causative for VO must be considered.

Therefore, the present study aims to answer the following questions. (1) What are the differences of HAVO and CAVO in terms of clinical presentation and treatment characteristics? (2) What are the causative pathogens in HAVO and CAVO? (3) What is the best possible empirical antibiotic therapy in both HAVO and CAVO? (4) Is there any difference of antibiotic susceptibility comparing pathogens evidenced in HAVO and CAVO?

## 2. Results

### 2.1. Patient Characteristics, Clinical Presentation, and Treatment (1)

Of the 253 patients with ICD-10 codes of interest, 155 were identified as having hematogenous vertebral osteomyelitis and met the inclusion criteria (Figure 1). In *n* = 65 cases, the above-mentioned criteria for the diagnosis of pyogenic VO were not fulfilled based on the medical records and, thus, they were excluded (incorrect ICD-10 coding). Table 1 displays the baseline data of all included patients and a differentiation between CAVO and HAVO-classified cases.

Overall, *n* = 88 (56.8%) of the patients were male and 67 (43.2%) female (male to female ratio = 1.3). The mean age was 66.1 ± 12.4 years. The mean body–mass index (BMI) was 29.0 ± 8.2 kg/m^2^. Most patients had comorbidities, resulting in a median Charlson comorbidity index (CCI) of 1 point (range 0–8 points; mean: 1.8 ± 1.8 points). Fifty-five (35.5%) patients showed a CCI between 2, 3, and 28 (18.1%), i.e., a CCI higher than three points. In 48 (31.0%) cases, the duration of symptoms could be determined on the basis of the medial records. The mean duration of symptoms was 63.7 ± 82.2 days. One-hundred and seven patients (69.0%) reported back pain at admission and *n* = 18 (11.6%) suffered from neurological deficits. Twenty-five (16.1%) patients presented with fever (body temperature > 38.0 °C). The mean first CRP (C-reactive protein) value was 132.4 ± 104.1 (range 3.0–498.0) mg/L and *n* = 66 (42.6%) patients had a CRP of >100 mg/L. 

There was no statistically significant difference in the mean CRP value between the CAVO (135.6 ± 111.7 mg/L) and the HAVO (128.5 ± 94.8 mg/L) cohort (*p* = 0.709). The mean leucocyte count at admission was 11.0 ± 4.6 (range 2.9–29.6) × 10^9^/L. Again, there was no difference in the leucocyte counts between the two cohorts (*p* = 0.094). During hospitalization, 34 (21.9%) patients developed septic symptoms. Twenty patients died during the hospitalization (in-hospital mortality rate = 12.9%). The most frequently documented reason for death was (septic) multi-organ failure (*n* = 10) followed by cardiac reasons (*n* = 4). In two cases, the reason for death was not documented in detail. There was no statistically significant difference in the number of in-hospital deaths between the CAVO (*n* = 9) and HAVO (*n* = 11) group (*p* = 0.485). The documented causes for in-hospital deaths are given in Table 2.

VO was diagnosed at the cervical spine in 12 (7.7%) cases, at the thoracic spine in 57 (36.8%) and at the lumbar spine in 74 (47.7%) cases (Table 1). A multifocal location of VO was seen in 12 (7.7%) patients.

According to the patient records, VO was classified as healthcare-associated (HA) in 74 (47.7%) of the cases and as community-acquired (CA) in 81 (52.3%). In all the HAVO-classified cases, the criterion (iii) was applicable, and, in 70 (94.6%) cases, a hospital admission within six months before symptom onset (ii) was documented.

There was no statistically significant difference in the age (*p* = 0.624), sex (*p* = 0.105), BMI (*p* = 0.390), and the mean CCI (*p* = 0.378) between CAVO and HAVO cases. There were no statistically significant differences in the incidence of fever (*p* = 0.367), b-symptoms (*p* = 0.174), or sepsis (*p* = 0.386) between the CAVO and HAVO cohort. The distribution of the localization of VO did not statistically or significantly differ between the CAVO and HAVO group (*p* = 0.195). 

Magnetic resonance imaging (MRI was conducted in 98 (63.2%) cases and a computerized tomography (CT) scan in 95 (61.3%) cases. In those, MRI revealed clear signs of VO in 99.0% of the scans and CT showed typical bone destruction in 98.9% of the scans. There was no significant difference in the number of cases with CT or MRI between the CAVO (CT *n* = 49 (60.5%) and MRI *n* = 51 (64.2%)) and the HAVO cohort (CT *n* = 46 (62.2%) and MRI *n* = 46 (62.2%)) *p* = 0.322 and *p* = 0.347). In the CAVO cohort, CT revealed signs of VO in 98.0% and MRI in 98.1%. In the HAVO cohort, both CT and MRI showed a sensitivity of 100.0%. In *n* = 42 (27.1%; CAVO; *n* = 25 (30.9%); HAVO; *n* = 17 (23.0%); *p* = 0.193) cases, a psoas abscess was identified, and paravertebral abscess formations were reported in *n* = 30 (19.4%; CAVO; *n* = 20 (24.7%); HAVO; *n* = 10 (13.5%); *p* = 0.064). An epidural abscess was found in *n* = 16 (10.3%; CAVO; *n* = 7 (8.6%); HAVO; *n* = 9 (12.2%); *p* = 0.321) cases.

Of all patients, 97 (62.6%) were treated surgically. In 28 (18.1%) cases, a CT-graphically guided abscess drainage was applied. Although HAVO patients showed a trend of longer duration of symptoms (59.6 ± 64.2 vs. 69.9 ± 96.9 days) and longer hospitalization time (32.3 ± 23.0 vs. 40.6 ± 46.5 days), those differences were not statistically significant (*p* = 0.588 and 0.157).

Ninety-eight patients (63.2%) received an empiric antibiotic treatment, prior to the identification of a pathogen. Flucloxacillin was administered most frequently in 24 (24.5%) cases, followed by a cephalosporin (ceftriaxon, cefazolin, or ceftazidim) in 18 cases (18.4%), and vancomycin (*n* = 13; 13.3%) and clindamycin in 7 cases (7.1%). Combined antibiosis was used in 70 cases (45.2%) and rifampicin was most frequently used for a combination in 24 cases (34.3%). From 45 cases, in which the antibiogram could be evaluated retrospectively, 26 patients received an empiric antibiotic treatment. The empiric antibiosis had to be changed accordingly to the antibiogram in 11 (42.3%) cases, whereas in 15 (57.7%) cases, the empiric antibiosis was found to be adequate.

### 2.2. Aetiology and Microbiological Pattern (2)

The most common isolate in the whole cohort was *S. aureus* (48.9%), coagulase-negative Staphylococci (CoNS) (26.1%), Enterobacteriales (10.2%), and Streptococcus species (6.8%). In 34 (42.0%) of the CAVO cases and in 33 cases (44.6%) of the HAVO patients, the infection was culture-negative. No polymicrobial infection was documented. Isolated microorganisms of both cohorts are presented in Figure 2. *S. aureus* was the most frequently detected pathogen in both cohorts (46.8% CAVO vs. 51.2% HAVO). This was followed by CoNS (21.3% CAVO vs. 31.7% HAVO; *p* = 0.440). Table 3 shows the distribution of causative pathogens between the HAVO and CAVO cohort with antibiograms; *S. aureus* was the most frequently detected pathogen (HAVO 45.5%; CAVO 47.8%). In the CAVO cohort, the second most frequently detected pathogens were Enterobactericales (21.7%), whereas in the HAVO cohort, CoNS were second most frequent (27.3%). However, this difference was not statistically significant (*p* = 0.342). Five of the six CoNS isolates and one *S. aureus* (MRSA) isolate in the HAVO group were methicillin-resistant (27.3%). In contrast, no methicillin-resistant isolates were found in the CAVO cohort (*p* < 0.01). In sum, 82.2% (CAVO: 73.9%; HAVO: 90.1%) of pathogens were classified as Gram-positive and 17.8% (CAVO: 26.1%; HAVO: 9.1%) were Gram-negative. No statistically significant difference in this distribution between the CAVO and the HAVO cohort was detected (*p* = 0.135).

### 2.3. Antimicrobial Regimes (3)

In both the CAVO and HAVO cohort, an antibiotic sensitivity of 100% could not be achieved with any mono-therapy. Only in the CAVO cohort, 100% sensitivity could be achieved by the combination of piperacillin–tazobactam and vancomycin or by the combination with meropenem and vancomycin. Figure 3 provides an overview of the predicted efficacy of empiric antimicrobial regimes in both cohorts.

In the HAVO cohort, the highest sensitivity was seen in a mono-therapies with teicoplanin (90.9%) and with linezolid (90.9%). The combination of meropenem plus vancomycin resulted in a high sensitivity of 95.5%. Both the combinations of vancomycin plus piperacillin–tazobactam or plus ceftriaxone resulted in a sensitivity of 90.9%. The combination of the reserve antibiotics teicoplanin or linezolid with piperacillin–tazobactam or ceftriaxone–cefotaxime–ceftazidim would reach a theoretical sensitivity of 100% each.

Overall, 82.2% (HAVO: 68.2%; CAVO: 95.5%) of the isolated pathogens were sensitive to piperacillin–tazobactam, 82.2% (HAVO: 90.9%, CAVO: 73.9%) to teicoplanin, 80.0% (HAVO: 90.9%; CAVO: 69.6%) to linezolid, 77.8% (HAVO: 81.8%; CAVO: 73.9%) to vancomycin, 64.4% (HAVO: 68.2%; CAVO: 60.1%) to clindamycin, and 55.6% (HAVO: 36.4%; CAVO: 73.9%) to ceftriaxone. Furthermore, 97.8% of pathogens were sensitive (HAVO: 95.5%; CAVO: 100.0%) to a combination of vancomycin plus meropenem, 91.1% were sensitive to vancomycin plus co-amoxiclav (HAVO: 86.4%; CAVO: 95.7%), 91.1% were sensitive to vancomycin plus ciprofloxacin (HAVO: 86.4%; CAVO: 95.7%), and 93.3% were sensitive to vancomycin plus cefotaxime (HAVO: 90.9%; CAVO: 95.7%). Teicoplanin or Linezolid plus piperacillin–tazobactam would reach a sensitivity of 100.0% (HAVO: 100.0%; CAVO: 100%). The lowest rates of resistance were evident for the combination of meropenem with vancomycin, for which only one strain remained resistant due to an infection with *S. epidermidis* (HAVO group). Highest overall rates of resistance were found for cefazolin (overall 34.8% resistant; HAVO: 45.5%; CAVO: 26.1%) and ciprofloxacin (overall 28.9% resistant; HAVO: 32.0%; CAVO: 26.1%), which could be reduced to 20.0% (HAVO: 18.2%; CAVO: 21.7%) and to 8.9% (HAVO: 13.6%; CAVO: 4.3%), respectively, by an additional combination with vancomycin.

The comparison of Gram-positive- and Gram-negative-stained pathogens showed differences in the distribution of resistances. In Gram-positive pathogens (*n* = 37 cases), the highest sensitivity rates were seen for vancomycin (94.6%), linezolid (97.3%), and teicoplanin (100%). The highest resistance rate was documented for ciprofloxacin (29.7%), followed by meropenem and cefazolin (both 24.3%) and piperacillin–tazobactam (21.6%; Figure 4A). In Gram-negative pathogens (*n* = 8 cases), the highest potential sensitivity was tested for meropenem, imipenem, cotrimoxazole, and piperacillin–tazobactam (each 100.0%). Besides vancomycin, linezolid, and teicoplanin, which are not considered suitable for the therapy of Gram-negative infections, the highest resistance rates were seen for cefazolin (87.5%) and ciprofloxacin (75.0%; Figure 4B). Comparing both groups, apart from vancomycin, linezolid, and teicoplanin (all *p* < 0.01), the only statistically significant difference in the rate of resistance was revealed for the therapy with cotrimoxazole (U = 88.000; Z = −2.146; *p* = 0.032).

### 2.4. Difference of Antibiotic Susceptibility of Pathogens Evidenced in HAVO and CAVO (4)

Comparing the predicted efficacy of empiric antimicrobial regimens between CAVO and HAVO patients revealed a statistically significant difference regarding the mono-therapy with meropenem (U = 172.000, Z = −2.654, *p* = 0.008), co-amoxiclav (U = 182.000, Z = −2.232, *p* = 0.026), piperacillin–tazobactam U = 183.500, Z = −2.382, *p* = 0.017), and ceftriaxone (U = 162.000, Z = −2.304, *p* = 0.021). The resistances seen in the HAVO cohort were 36.4% for meropenem, 31.8% for co-amoxiclav, 31.8% for piperacillin–tazobactam, and 27.3% for ceftriaxone. No statistically significant differences were found between CAVO and HAVO patients regarding tested combined antibiotics (Supplemental Material Appendix A).

## 3. Discussion

The present study compares clinical features and microbiological epidemiology of healthcare-associated and community-acquired vertebral osteomyelitis. Based on antibiotic susceptibility testing of the isolated pathogens, a variety of treatment regimens for empirical antibiotic treatment were analyzed and the most efficient choices were evaluated. The results revealed that underlying pathogens were not statistically significant different for HAVO and CAVO. However, best practice options regarding the sensitivity of antimicrobial regimes were distinct. Antibiotic combinations of either piperacillin–tazobactam + vancomycin or meropenem + vancomycin seem reasonable to achieve a 100% safe susceptibility to empirical antibiotic therapy for CAVO and approximately 90% for HAVO.

### 3.1. Patient Characteristics and Clinical Presentation and Treatment (1)

A total of 155 patients with a male to female ratio of 1.3 and a mean age of 66.1 years were included in this study. This is in line with epidemiological studies on VO [1,2,14]. In the current cohort, consisting of 47.7% HAVO and 52.3% CAVO cases, no significant differences in the patient details (age, sex, BMI) or presented symptoms were identified. In contrast, it has been reported that elderly patients had higher rates of healthcare-associated vertebral osteomyelitis than younger patients [15]. Noteworthy, the median CCI was by tendency higher in the HAVO group, and the mean duration of symptoms was around ten days longer compared to the CAVO group. Remarkably, only 25 (16.1%) patients presented with fever. Despite this, the mean first CRP value was 132.4 ± 104.1 mg/L and *n* = 66 (42.6%) patients had a CRP of >100 mg/dL. HAVO was associated with a longer duration of symptoms and a longer stay in hospital. 

The overall in-hospital mortality rate was 12.9%, without a relevant difference between the groups (CAVO 11.1%; HAVO 14.9%). In a current prospective study, Yagdiran reported a 1- and 2-year mortality rate of 20% and 23%, respectively [16]. They exclusively followed surgically treated patients, whereas in our study, we did not differ between the conservatively and surgically treated patients and only reported on the in-hospital mortality. Pigrau et al. described the risk factors, infectious sources, etiology, clinical features, therapy, and outcome of hematogenous HAVO, and compared the findings with those of CAVO cases in a retrospective cohort study including 41 HAVO cases. They found that, in their setting, one-third of hematogenous pyogenic VO infections were healthcare-associated [9]. Compared with CAVO, the mortality and relapse rates were higher. This is in line with the presented findings that, even though they are not statistically significant. Our findings and the current literature underline that VO, despite advances in treatment options, is still a severe condition associated with high mortality. The influence of a healthcare-associated infection as a potential predictive factor for the survival and clinical outcome remains to be evaluated.

HAVO patients showed a trend of longer duration and longer hospitalization time than CAVO patients, but without statistical significance. Moreover, 63.2% of patients received an empiric antibiotically treatment, prior to the identification of a pathogen. The empiric antibiosis was found to be inadequate in 42.3% of cases with an antibiogram. It is crucial that empiric antibiotic should be based on the host and the epidemiologic risk, as well as the local susceptibility patterns. Culture-negative VO is of great importance, in particular. In a retrospective study on 73 patients, Yu et al. recently reported that β-lactam and glycopeptide antibiotics were mainly used in VO, and fluoroquinolones were used statistically and significantly more in culture-negative VO [17]. There are limited evidence-based data on national recommendation for the empiric antibiotic regime in VO. There might be considerable differences in daily clinical practice due to a high heterogeneity in local guidelines and experiences that remain to be illuminated more precisely.

### 3.2. Aetiology and Microbiological Pattern (2)

VO is often difficult to diagnose, especially considering the high rate of patients on antibiotic therapy at the time of diagnosis. This observation results in a low rate of positivity from blood cultures and vertebral biopsy to determine the etiology of infection, causing a high rate of culture-negative cases (43.2%). Even though it is suggested that antibiotic therapy should be withheld until the pathogens are identified, this is often not feasible in clinical practice when patients’ conditions are unstable [13].

In a 10-year cohort study, Avenel et al. recently found that, when VO was suspected on imaging, bacteriological investigation identified the microorganism in 209/300 (70%) of the cases [18]. The yield of percutaneous needle biopsy was 54.8% and the only predictor of percutaneous needle biopsy negativity, detected in a multivariate analysis was previous antibiotic intake [18]. There was no statistically significant difference in the frequencies of causative pathogens between the HAVO and CAVO cohort. For both cohorts, *S. aureus* was the most frequently detected pathogen in both subgroups, which is in line with the literature [6,17,19]. The analyzed cohorts did not include polymicrobial infections, which is also in line with other studies reporting low percentages [20]. In the present cohort, 82.2% of causative pathogens were classified as Gram-positive and 17.8% as Gram-negative, with no significant differences between the CAVO and HAVO cohort. We found *S. aureus* to be the most frequent and CoNS to be the second most frequent causative pathogen in the HAVO cohort.

In their study, which included 358 VO cases, Park et al. found that methicillin-susceptible *S. aureus* was the most frequently isolated causative pathogen (33.5%), followed by methicillin-resistant *S. aureus* (MRSA) (24.9%), Enterobacteriales (19.3%), and Streptococcus species (11.7%) [21]. Concordant with our study, they did not find differences in the proportions of pathogens between the HAVO and non-HAVO cohort. They called attention to the higher frequency of MRSA isolates in the HAVO group (43.6% vs. 13.8%), whereas MSSA and Streptococcus spp. were more often isolated in non-HAVO cases [21]. In contrast, we only found one MRSA isolate in the HAVO group but substantial differences in the distribution of methicillin-resistant and healthcare-associated CoNS. Pigrau et al. documented a CoNS infection in 15.0% in HAVO and of 2.0% in CAVO cases [9]. A similar trend was seen in our results, which show infections with CoNS in 31.7% of HAVO and in 21.3% of CAVO cases. Significantly more methicillin-resistant isolates were found in the HAVO cohort compared to CAVO. In their update on the increasing clinical impact of CoNS infections, Michels et al. point out the extensive antimicrobial resistance profile of CoNS, especially in healthcare settings [22]. Because of the lower virulency of CoNS compared to *S. aureus* in VO [23], these infections may be missed and diagnosed late. Over the past few years, CoNS infections, usually originating from intravascular devices, have been increasingly reported as a cause for hematogenous VO [12,24,25]. Intravascular catheter-associated infections are considered as the sources of healthcare-associated vertebral osteomyelitis in up to 34% of patients [9]. The higher incidence of antibiotic resistance in causative pathogens, predominantly in CoNS in the current HAVO cohort, must be considered when selecting an empiric antibiotic regimen. We showed that the patients in the HAVO cohort are, by tendency, older with more comorbidities, according to the CCI. Although not directly proven by our results, it can be hypothesized that older and multi-morbid patients are more prone to infections with resistant CoNS, not least because of frequent medical interventions.

### 3.3. Empirical Antibiotic Therapy Regimes and Differences between CAVO and HAVO (3 and 4)

Antibiotic therapy is essential for the treatment of VO. However, guidelines targeting antibiotic treatment strategies are scarce and only limited data are available on the efficacy of antibiotics [26,27]. For the selection of an appropriate empiric antibiotic regime, the local antibiogram profiles should be respected. The highest overall rates of resistance were found for cefazolin (overall 34.8% resistant; HAVO: 45.5%; CAVO: 26.1%) and ciprofloxacin (overall 28.9% resistant; HAVO: 32.0%; CAVO: 26.1%). A statistically significant difference regarding the mono-therapy between HAVO and CAVO was seen for meropenem, co-amoxiclav, piperacillin–tazobactam and ceftriaxone. Considering the high rates of resistance for antibiotic mono-therapies both in CAVO and HAVO patients, these must be critically considered in VO.

The main proportion of vancomycin resistance in this study cohort can be lead back to Gram-negative pathogens. Furthermore, a Gram-negative pathogen was only identified in 17.8% of cases, and there was no significant difference in the distribution between the HAVO and the CAVO cohort. Similarly, Park et al. found a rate of 20.8% for Gram-negative bacteria in their cohort of 313 microbiologically diagnosed VO cases [28]. By trend, more Gram-negative strains were found in the current CAVO cohort. They showed sensitivities of 100% for the testes carbapenems, cotrimoxazole, and piperacillin–tazobactam.

Our results indicate a broad coverage for either piperacillin–tazobactam + vancomycin, vancomycin + co/amoxiclav, vancomycin + ciprofloxacin, vancomycin+ ceftotaxime, or meropenem + vancomycin, achieving a 95% safe susceptibility to empirical antibiotic therapy for CAVO and more than 80% for HAVO. The highest potential coverage in HAVO cases of 95.5% could be achieved with vancomycin + meropenem. These combinations are most likely to cover Gram-positive and Gram-negative stains. The use of carbabenems should, however, be limited to critically ill patients since the use of carbapenems has been proven to be associated with the risk of acquiring antibiotic-resistant bacteria colonization [29]. A broad-spectrum antibiotic combination encompasses the risk of facilitating antimicrobial resistance. Thus, antimicrobial susceptibility assessment is essential, and initial antibiotic therapy should rapidly be adjusted as soon as pathogens and their antibiograms are identified. 

Only a few studies focused on the difference between CAVO and HAVO. Park et al. found that most of the isolated pathogens were susceptible to vancomycin plus ciprofloxacin or broad-spectrum cephalosporin for HAVO [21]. Their findings further suggested that fluoroquinolone-based oral combinations may not be appropriate due to frequent resistance, especially in cases of HAVO [21]. In their cohort, MRSA was the main causative pathogen, whereas in our cohort, only one MRSA case was evident. In contrast, methicillin resistance was mainly found in CoNS isolates in the current study. This difference highlights the importance of the evaluation of local susceptibility patterns.

For critically ill patients and patients with renal dysfunction, the vancomycin dosage must be adapted, and regular control of the serum levels are essential to prevent nephrotoxicity. As alternatives to vancomycin, teicoplanin, linezolid, and daptomycin can be considered. The susceptibility pattern for daptomycin was not routinely tested in our institution; hence, the present study can unfortunately not provide any data on this agent. However, a retrospective analysis of MRSA vertebral osteomyelitis cases showed a higher recurrence rate under vancomycin compared to daptomycin [30]. Linezolid and teicoplanin showed high susceptibility rates (each 90.9%) when used as a mono-therapy in HAVO cases but substantial lower rates in CAVO cases (69.6% and 73.9%, respectively). If combined with piperacillin–tazobactam, linezolid and teicoplanin both showed a theoretical sensitivity of 100%, both in the HAVO and the CAVO cohort. Another feasible approach to bypass unwanted side effects of systemic antibiotics is via the administration of local antibiotic carriers in cases of implant-associated vertebral osteomyelitis [27]. For instance, gentamicin + vancomycin, which should be carefully considered as systemic antibiotic therapy due to nephrotoxicity, are applicable as local antibiotics. Regarding this, Fleege et al. showed a reduction in the duration of systemic antibiotic therapy by the additional use of local antibiotics based on absorbable carrier material in the surgical treatment of bacterial VO [31]. Their study reveals that gentamicin and vancomycin are commonly used agents in local carrier materials. In our cohort, 86.4% of isolates in the HAVO cohort showed a theoretical susceptibility for a combination of gentamicin + vancomycin, but only 59.1% of isolates were susceptible to gentamicin alone. 

### 3.4. Limitations

This retrospective study has several limitations. Data analysis of one orthopaedic-trauma center may lead to a local epidemiological bias. In addition, the retrospective design restricts analysis to already-existing resistograms, which were electronically available back to 2013. In some cases, antibiotic testing for certain antibiotics was sometimes not performed, leading to “unknown” listed antibiotic susceptibility. In addition, the retrospective file analysis did not consistently allow identification of antibiotic pretreatment and its effect on the detection of infection-causing pathogens.

## 4. Materials and Methods

### 4.1. Patient Identification

A retrospective cohort study of patients treated for VO was conducted in a level-1 orthopaedic trauma center with a dedicated spine focus in Germany. The inclusion period was defined from 1 January 2000 to 3 December 2020. Eligible cases of patients 18 years or older were screened by international classification of disease (ICD)-10 diagnosis codes (M46.2: osteomyelitis of vertebra; M46.3: infection of the intervertebral disc (pyogenic); M46.4: discitis, unspecified; and M46.5: other infective spondylopathies). Afterwards, patients’ medical charts, surgery protocols, laboratory findings, as well as microbiological and histopathological reports were screened for criteria of VO. The diagnosis of VO was confirmed if at least two of the following criteria were documented: Compatible clinical features (1); radiological evidence of vertebral osteomyelitis in CT and/or MRI (2) [32,33]; and microbiologic demonstration of bacterial pathogens, either from the site of infection itself (e.g., abscess, intervertebral disc, or vertebral bone) or in the blood (3) [6,34]. As previously defined by Pigrau et al., HAVO was assumed when at least one of the following criteria were documented [9]:(i)Onset of symptoms after one month of hospitalization with no evidence of vertebral osteomyelitis at admission;(ii)Hospital admission within six months before symptom onset;(iii)Ambulatory diagnostic or therapeutic manipulations within six months before symptom onset (long-term central venous catheter use, arteriovenous fistula for hemodialysis, invasive intravascular techniques, urological, gynecological or digestive procedures, and cutaneous manipulations).

VO cases that did not meet any of the above criteria were classified as CAVO.

### 4.2. Data Collection

Patient characteristics (sex, age, and BMI at the time of admission), symptoms (backpain, neurological symptoms, fever, and sepsis), clinical features (start of symptoms, hospitalization, treatment modalities, and empiric antibiotics), blood parameters (CRP and leucocyte count), radiological findings (signs of vertebral osteomyelitis in the CT or MRI), and details of VO (spinal height, vertebral levels, and abscess formations (psoas, paravertebral, epidural)) were assessed retrospectively by reviewing electronic medical records. Comorbidities were assessed by obtaining the CCI [35]. The microbiological database was searched for information on the causative pathogens and on their antimicrobial susceptibility testing. Detection was achieved either preoperatively or intraoperatively by deep tissue sampling or aspiration of the affected vertebral segment. Data on antimicrobial susceptibility testing were accessible back to 2013.

### 4.3. Microbiology

Tissue samples were homogenized and seeded on solid and liquid culture media. All samples were incubated for 14 days. Bacteria were identified by matrix-assisted laser desorption ionization time of flight mass spectrometry (MALDI TOF MS) using a Microflex LT mass spectrometer and BioTyper software (Bruker Daltonik, Bremen, Germany). Antibiotic susceptibility testing followed guidelines from the European Committee on Antimicrobial Susceptibility Testing (EUCAST) were applied [36]. Testing was performed using a BD Pheonix M50 nephelometer (Becton, Dickinson and Company, Heidelberg, Germany) or manually by disc diffusion. 

### 4.4. Statistics

Descriptive and statistical data analysis was performed using the IBM SPSS Statistics software (version 28.0, IBM Corp, Armonk, NY, USA). Frequencies were expressed as numbers and percentages. Continuous parameters were presented as means ± standard deviation (SD) and compared by Student’s *t*-test. The Chi-square test was used for comparison of categorical variables. The Mann–Whitney-U-test was used to determine if there were differences in the antimicrobial regimes between CAVO and HAVO patients or between cases with Gram-positive and Gram-negative pathogens. For all tests, *p* values < 0.05 were considered statistically significant.

## 5. Conclusions

Healthcare association is common in VO. Antibiotic resistance of CAVO pathogens differed significantly from HAVO. We detected significantly more methicillin-resistant isolates in the HAVO cohort compared to the CAVO cohort. The hypothetical analysis of possible antibiotic regimes reveals that, for an empirical antibiotic therapy, a combination of β-lactam antibiotics + vancomycin is suitable, both for CAVO and HAVO cases.

## Figures and Tables

**Figure 1 antibiotics-10-01410-f001:**
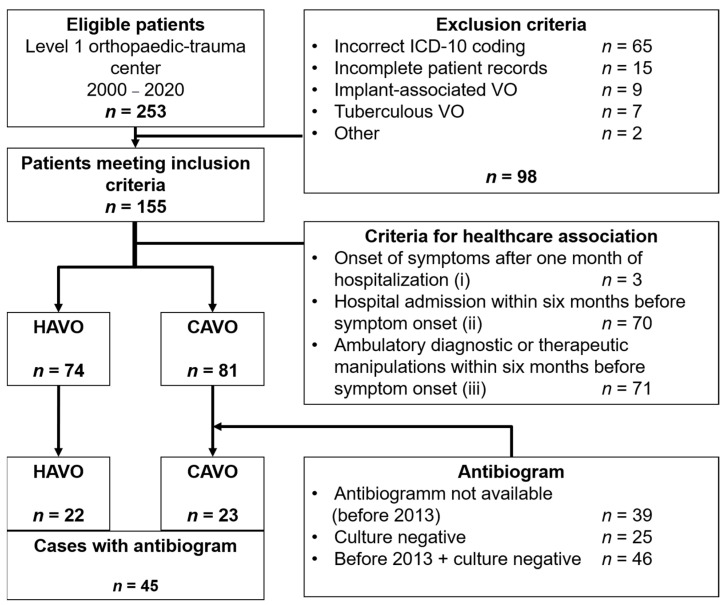
Flowchart of patients included in the study. Abbreviations: HAVO: Healthcare-associated vertebral osteomyelitis. CAVO: Community-acquired osteomyelitis.

**Figure 2 antibiotics-10-01410-f002:**
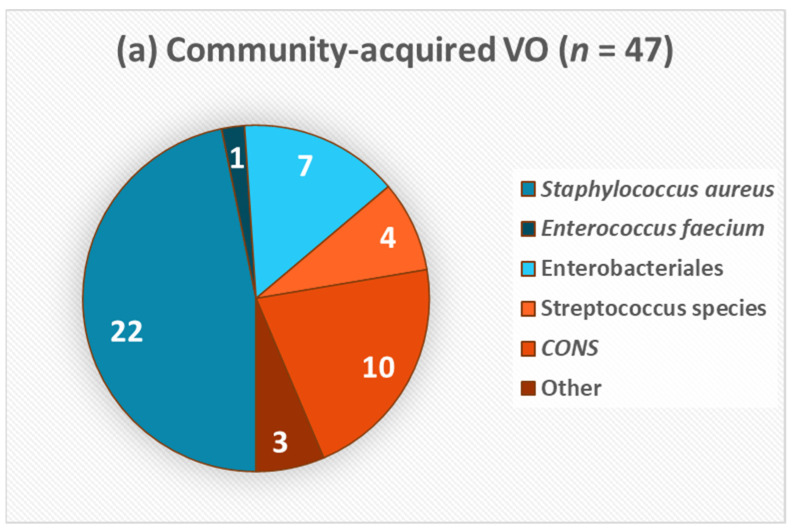
Isolated microorganisms shown in percentage from (**a**) CAVO and (**b**) HAVO patients. Other pathogens in CAVO include Propionibacterium acnes, Haemophilus parainfluenzae, and Cutibacterium avidum.

**Figure 3 antibiotics-10-01410-f003:**
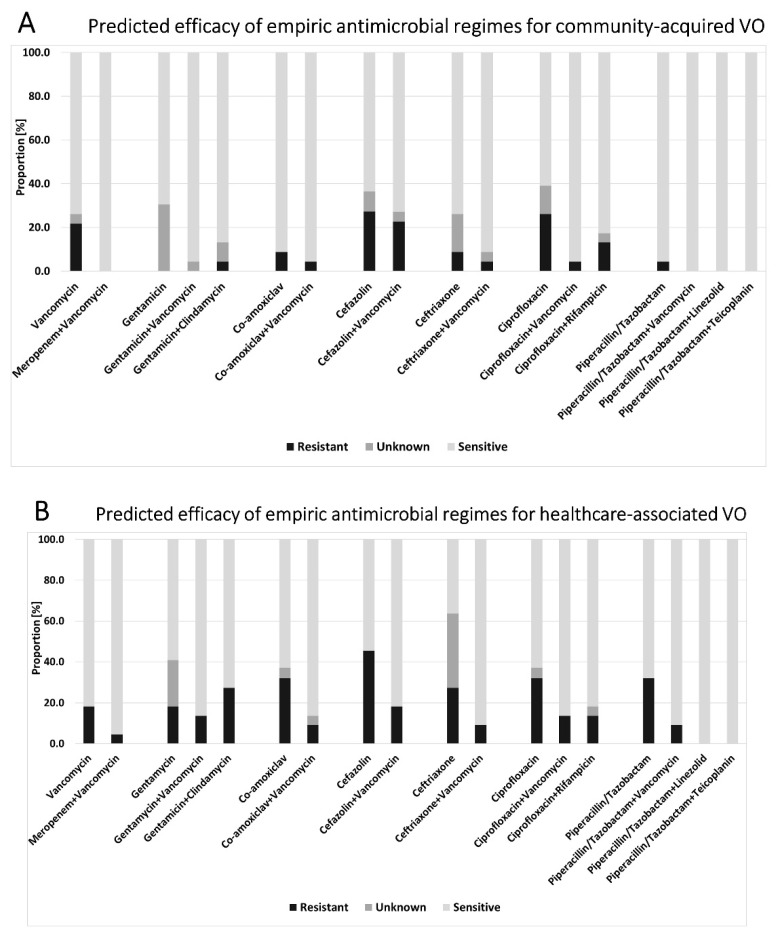
Predicted efficacy of empiric antimicrobial regimes for (**A**) the CAVO cohort and (**B**) the HAVO cohort. Bars represent cumulative proportions of resistant and sensitive pathogens and pathogens with unknown sensitivity against the respective agent.

**Figure 4 antibiotics-10-01410-f004:**
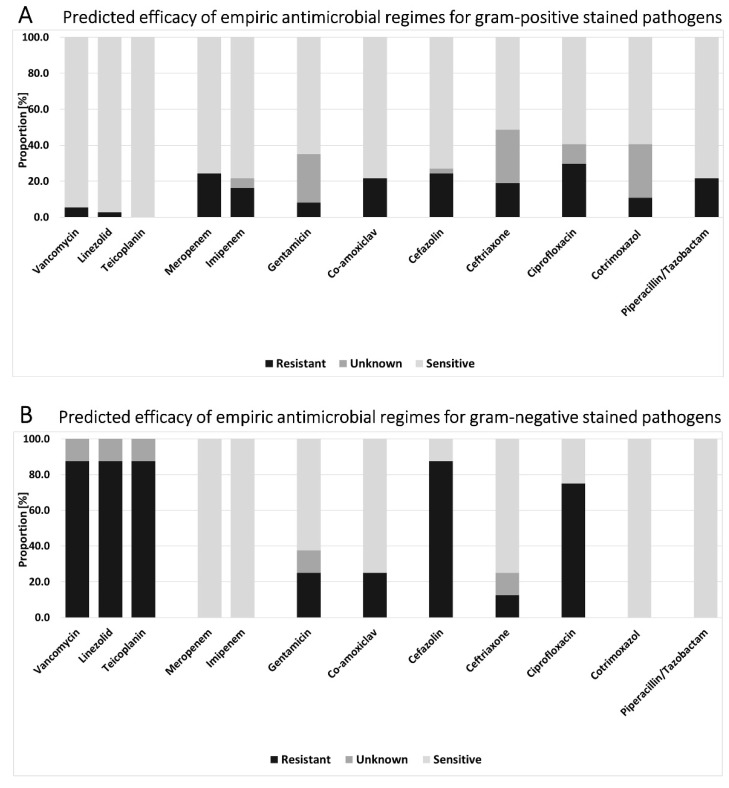
Predicted efficacy of empiric antimicrobial regimes for (**A**) Gram-positive-stained pathogens (*n* = 37) and (**B**) Gram-negative-stained pathogens (*n* = 8). Bars represent cumulative proportions of resistant and sensitive pathogens as well as pathogens with unknown sensitivity against the respective agent.

**Table 1 antibiotics-10-01410-t001:** Baseline characteristics of all patients and the two cohorts consisting of CAVO and HAVO patients. Data on age, BMI, duration of symptoms, and hospitalization is given as mean values ± standard deviation. CCI points are given as median with range and mean ± standard deviation. All other parameters are displayed numerically and as a percentage of the respective cohort.

Characteristic	All (*n* = 155)	CAVO (*n* = 81)	HAVO (*n* = 74)
Demographic data			
Sex (male)	88 (56.8%)	41 (50.6%)	47 (63.5%)
Age (years)	66.1 ± 12.4	66.6 ± 12.9	65.6 ± 11.9
BMI (kg/m^2^)	29.0 ± 8.2	29.9 ± 9.0	28.4 ± 7.7
CCI	1 [0–8] (1.8 ± 1.8)	1 [0–7] (1.5 ± 1.6)	2 [0–8] (2.1 ± 1.9)
Location			
Cervical spine	12 (7.7%)	8 (9.9%)	4 (5.4%)
Thoracic spine	57 (36.8%)	34 (42.0%)	23 (31.1%)
Lumbar spine	74 (47.7%)	35 (43.2%)	39 (52.7%)
Multifocal	12 (7.7%)	4 (4.9%)	8 (10.8%)
Duration of symptoms (days)	63.7 ± 82.2	56.9 ± 64.2	69.9 ± 96.9
Hospitalization (days)	36.3 ± 36.3	32.3 ± 23.0	40.6 ± 46.5
In-hospital deaths	20 (12.9%)	9 (11.1%)	11 (14.9%)
Microbiologic results			
Culture-negative	67 (43.2%)	34 (41.9%)	33 (44.6%)
Antibiogram available	45 (29.0%)	23 (28.4%)	22 (29.7%)
Positive blood culture	70 (45.2%)	35 (43.2%)	35 (47.3%)

**Table 2 antibiotics-10-01410-t002:** Causes of in-hospital deaths in the CAVO and the HAVO cohort in total numbers and percentages of cases. The age at the time of death is given separately for each case and as mean ± standard deviation for the respective cohort.

	CAVO (*n* = 9)			HAVO (*n* = 11)		
Reason	*n*	%	Age [years]	*n*	%	Age [years]
Cardiovascular arrest	2	22.2	65; 74	2	18.2	66; 79
Multi-organ failure	3	33.3	69; 75; 77	4	36.4	50; 71; 71; 74
Sepsis (with multi-organ failure)	3	33.3	74; 83; 83	-	-	-
Drug intoxication	1	11.1	32	-	-	-
Respiratory insufficiency	-	-	-	1	9.1	69
Graft versus host disease	-	-	-	1	9.1	67
Fungal pneumonia	-	-	-	1	9.1	63
Not documented	-	-	-	2	18.2	79; 85
	Mean age		70.1 ± 15.4	Mean age		70.4 ± 9.4

**Table 3 antibiotics-10-01410-t003:** Isolated microorganisms in CAVO and HAVO patients with antibiogram (*n* = 45).

Pathogen	CAVO (*n* = 23)	HAVO (*n* = 22)
*Staphylococcus aureus*	11 (47.8%)	10 (45.5%)
Coagulase-negative staphylococci	2 (8.7%)	6 (27.3%)
Streptococcus species	2 (8.7%)	2 (9.1%)
Enterobactericales	5 (21.7%)	2 (9.1%)
Enterococcus species	1 (4.3%) (*E. faecium*)	2 (9.1%)
Other	2 (8.7%) (*Haemophilus parainfluenzae; Cutibacterium avidum*)	-

## Data Availability

All data presented in this study were available on demand from the corresponding author.

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
