# Peer review of "Is There a Difference in Clinical Features, Microbiological Epidemiology and Effective Empiric Antimicrobial Therapy Comparing Healthcare-Associated and Community-Acquired Vertebral Osteomyelitis?"

_antibiotics, 2021, doi:10.3390/antibiotics10111410_

Round 1

Reviewer 1 Report

The present study retrospectively compared clinical features and microbiological epidemiology of HAVO and CAVO. Also, effective empiric antimicrobial therapy and antimicrobial coverage were analyzed. Accordingly, this data based results was able to provide a valuable guideline for clinical antibiotic application.     

Many in brief terms such as CCI, CRP and BMI et al should give full name when were first indicated. Also, CCI standard was wish to be given in your manuscript.

In your result section, “Twenty patients died during the hospitalization (mortality rate = 12.9%).” If possible, the more detail description on those patients is wish to be list in a table, which could be give readers more valuable information on those issue.

“MRI was conducted in 98 (63.2%) cases and a CT scan in 95 (61.3%) cases.”… in this paragraph, authors neglect a description of cases in receptively two group and a comparation between them.

The colors applied in your figure 2 were similar. Could authors readjust them and make them more distinguishable? 

The results section mentioned “In the HAVO cohort no 100.0% sensitivity could be theoretically achieved with first-line antibiotics”. What is your definition on first-line antibiotics?

Teicoplanin or Linezolid plus piperacillin/tazobactam would reach a sensitivity of 100.0% (HAVO: 100.0%, CAVO: 100.0%). Why this combination result was not noted in your figure 3

In discussion section, line 248 “…is still is a severe condition associated…” seems duplicated a is, please check.

Author Response

Dear editor, dear reviewers,

Thank you very much for providing us with the overall positive comments of the peer-reviewers on our proposed manuscript entitled “Is there a difference in clinical features, microbiological epi-demiology and effective empiric antimicrobial therapy com-paring healthcare-associated and community-acquired vertebral osteomyelitis?”.

We very much appreciate the opportunity to revise our manuscript, and we would like to inform you that we were able to respond to all the reviewers' comments. Point by point responses were uploaded. We marked all new changes in the revised manuscript by using the word track change mode. Each major change was also presented to the reviewers in the answer statements files. We ask for your understanding that we have also made some minor textual changes above that.

We believe we were able to further improve the quality of our manuscript based on the valuable comments and suggestions of the reviewers and are thankful for this support.

If any further questions or any dubiety appears, please do not hesitate to contact us.

Reviewer 1:

The present study retrospectively compared clinical features and microbiological epidemiology of HAVO and CAVO. Also, effective empiric antimicrobial therapy and antimicrobial coverage were analyzed. Accordingly, this data based results was able to provide a valuable guideline for clinical antibiotic application.     

Many in brief terms such as CCI, CRP and BMI et al should give full name when were first indicated. Also, CCI standard was wish to be given in your manuscript.

  • Thank you pointing out this issue. We apologize for the flaw and made sure, that all abbreviations are given with the full name when first mentioned in the text. We added the mean ± standard deviation values fort the CCI. We assumed it would not add to the informative value of the manuscript to give the detailed explanation of the contributing diseases of the CCI. The citation of the original publication is stated in line 449 [35].

In your result section, “Twenty patients died during the hospitalization (mortality rate = 12.9%).” If possible, the more detail description on those patients is wish to be list in a table, which could be give readers more valuable information on those issue.

  • Thank you for this valuable suggestion. Accordingly, we added a new table (Table 2) that gives an overview on the causes of in hospital deaths for the CAVO and the HAVO cohort. We also listed the age of the patients and gave a mean age at deaths. Unfortunately, due to the retrospective character of this study and the dependency of quality of documentation we were not able to comprehend the cause of death in 2 cases and classified them as “not documented”.

 “MRI was conducted in 98 (63.2%) cases and a CT scan in 95 (61.3%) cases.”… in this paragraph, authors neglect a description of cases in receptively two group and a comparation between them.

  • Thank you for pointing this out. The intention in not to explicitly compare both groups in the radiology paragraph was not to overload the manuscript, as we did not consider this information essential for the main questions. However, we, agree that the comparison of the occurrence of abscesses is important for the description of the clinical presentation. As suggested, we supplemented the paragraph with the comparison of both groups:

“There was no significant difference in the number of cases with CT or MRI between the CAVO (CT n=49 (60.5%) and MRI n=51 (64.2%)) and the HAVO cohort (CT n=46 (62.2%) and MRI n=46 (62.2%)) p=0.322 and p=0.347). In the CAVO cohort CT revealed signs of VO in 98.0% and MRI in 98.1%. In the HAVO cohort both CT and MRI showed a sensitivity of 100.0%. In n=42 (27.1%; CAVO; n=25 (30.9%); HAVO; n=17 (23.0%); p=0.193) cases a psoas abscess was identified, and paravertebral abscess formations were reported in n=30 (19.4%; CAVO; n=20 (24.7%); HAVO; n=10 (13.5%); p=0.064). An epidural abscess was found in n=16 (10.3%; CAVO; n=7 (8.6%); HAVO; n=9 (12.2%); p=0.321) cases.” (Lines 132-143)

 The colors applied in your figure 2 were similar. Could authors readjust them and make them more distinguishable? 

  • Thank you for this suggestion. We readjusted the colors and hope to have improved the distinguishability and contrast.

 The results section mentioned “In the HAVO cohort no 100.0% sensitivity could be theoretically achieved with first-line antibiotics”. What is your definition on first-line antibiotics?

  • Thank you for asking for the definition of first-line antibiotics. We tend to define them as agents, that commonly not serve as a substitute and are reasonable used as a first therapy attempt, often as empiric antibiotics. As thematized in lines 388-390 we discussed linezolid and teicoplanin to be alternatives for vancomycin. Nevertheless, the definition might be controversial, and it does not add to the informative value of the results to make them depend from this definition. Further, there was no 100.0% sensitivity for any mono-therapy in the HAVO cohort, so we decided to delete the sentence. We hope you find this acceptable. The new sentence reads: “In the HAVO cohort the highest sensitivity was seen in a mono-therapies teicoplanin (90.9%) and with linezolid (90.9%).” (Lines 194-196)

 Teicoplanin or Linezolid plus piperacillin/tazobactam would reach a sensitivity of 100.0% (HAVO: 100.0%, CAVO: 100.0%). Why this combination result was not noted in your figure 3

  • Thank you very much for this question. As discussed above, we do not consider linzeolid and Teicoplanin as antibiotics that are usually chosen for empiric antibiotic treatments. However, we agree, that the information in the text should be supported by the graphical presentation. Therefore we added the combinations piperacillin/tazobactam + linezolid and + teicoplanin to figures 3A and B.

 In discussion section, line 248 “…is still is a severe condition associated…” seems duplicated a is, please check.

  • Thank you for pointing this out. We corrected it. (New line 289)

Reviewer 2 Report

This is an interesting study comparing the microbiological profile (aetiology, antibiotic resistance) between patients with community and health-care associated vertebral osteomyelitis. A few points to be addressed by the authors before this paper is finally accepted:

(a) Material and methods: (i) please add references for the methods employed for the identification of bacterial strains and for the EUCAST guidelines (ii) please add the method employed in the antibiotic resistance testing 

 (b) Figure 2: please correct Enterobacterales to Enterobacteriales, Pseudomonas aeuriginosa to Pseudomonas aeruginosa. Please also correct the names throughout the text. 

(c) The presentation of antibiotic resistance is confusing since different antibiotics are tested for gram (+) bacteria such as Staphylococcus and Enterococcus sp and different for gram (-). Please present the antibiotic resistance for gram (+) and gram (-) separately. 

Author Response

Dear editor, dear reviewers,

Thank you very much for providing us with the overall positive comments of the peer-reviewers on our proposed manuscript entitled “Is there a difference in clinical features, microbiological epi-demiology and effective empiric antimicrobial therapy com-paring healthcare-associated and community-acquired vertebral osteomyelitis?”.

We very much appreciate the opportunity to revise our manuscript, and we would like to inform you that we were able to respond to all the reviewers' comments. Point by point responses were uploaded. We marked all new changes in the revised manuscript by using the word track change mode. Each major change was also presented to the reviewers in the answer statements files. We ask for your understanding that we have also made some minor textual changes above that.

We believe we were able to further improve the quality of our manuscript based on the valuable comments and suggestions of the reviewers and are thankful for this support.

If any further questions or any dubiety appears, please do not hesitate to contact us.

This is an interesting study comparing the microbiological profile (aetiology, antibiotic resistance) between patients with community and health-care associated vertebral osteomyelitis. A few points to be addressed by the authors before this paper is finally accepted:

(a) Material and methods: (i) please add references for the methods employed for the identification of bacterial strains and for the EUCAST guidelines (ii) please add the method employed in the antibiotic resistance testing 

  • Thank you very much for this important suggestion. We added the source for the EUCAST guidelines (i): EUCAST: AST of bacteria n.d. https://www.eucast.org/ast_of_bacteria/ (accessed November 8, 2021).

Further we have supplemented the description of the susceptibility testing (ii): “Testing was performed using a BD Pheonix M50 nephelometer or manually by disc diffusion.” (Lines 460-461)

 (b) Figure 2: please correct Enterobacterales to Enterobacteriales, Pseudomonas aeuriginosa to Pseudomonas aeruginosa. Please also correct the names throughout the text. 

  • Thank you for this comment. We apologize for these typos and corrected all names in the figures and throughout the text as suggested.

(c) The presentation of antibiotic resistance is confusing since different antibiotics are tested for gram (+) bacteria such as Staphylococcus and Enterococcus sp and different for gram (-). Please present the antibiotic resistance for gram (+) and gram (-) separately. 

  • Thank you for this important remark. The current retrospective evaluated population showed mainly gram (+) bacteria as causative pathogens in 82.2% of cases, whereas gram (-) pathogens (E. coli, Salmonella mikawasima, Haemophilus parainfluenzae, Klebsiella pneumoniae) were seen in only 17.8% of cases. We supplemented the results section with this interesting new finding:

“In sum 82.2% (CAVO: 73.9%; HAVO: 90.1%) of pathogens were classified gram positive and 17.8% (CAVO: 26.1%; HAVO: 9.1%) gram negative. No statistically significant differ-ence in this distribution between the CAVO and the HAVO cohort was detected (p=0.135).” (Lines 173-176)

We agree, that tested antibiotics usually differ according to the gram staining of the pathogen. However, the used database provides information on the resistances for all pathogens and all presented antibiotics, except for clindamycin. We added the figures 4A and B to provide an overview on differences in the resistance profile between gram (+) and (-) pathogens. We also identified a statistically significant difference for the resistance rate of cotrimoxazole. Notable the statistical testing must be considered carefully because of the in-homogenous sample sizes (n=37 vs. n=8). To address these findings we added the following paragraphs to the results and discussion section respectively:

“The comparison of gram-positive and gram-negative stained pathogens showed differ-ences in the distribution of resistances: In gram-positive pathogens (n=37 cases) the high-est sensitivity rates were seen for vancomycin (94.6%), linezolid (97.3%) and teicoplanin (100.0%). The highest resistance rate was documented for ciprofloxacin (29.7%), followed by meropenem and cefazolin (both 24.3%) and piperacillin/tazobactam (21.6%; Figure 4A). In gram-negative pathogens (n=8 cases) the highest potential sensitivity was tested for meropenem, imipenem, cotrimoxazol and piperacillin/tazobactam (each 100.0%). Be-sides vancomycin, linezolid and teicoplanin, which are not considered suitable for the therapy of gram-negative infections the highest resistance rates were seen for cefazolin (87.5%) and ciprofloxacin (75.0%; Figure 4B). Comparing both groups, apart from vanco-mycin, linezolid and teicoplanin (all p<0.01) the only statistically significant difference in the rate of resistance was revealed for the therapy with cotrimoxazole (U=88.000; Z=-2.146; p=0.032).” (Lines 226-238)

“In the present cohort in 82.2% of causative pathogens were classified gram positive and 17.8% as gram negative with no significant differences between the CAVO and HAVO cohort. We found Staph. aureus to be the most frequent and CoNS to be the second most frequent causative pathogen in the HAVO cohort.” (Lines 320-323)

“The main proportion of vancomycin resistance in this study cohort can be led back to gram-negative pathogens. Anyways, only in 17.8% of cases a gram-negative pathogen was identified and there was no significant difference in the distribution between the HAVO and the CAVO cohort. Similar Park et al. found a rate of 20.8% for gram-negative bacteria in their cohort of 313 microbiologically diagnosed VO cases [28]. By trend more gram-negative strains were found in the current CAVO cohort. Those showed sensitivities of 100.0% for the testes carbapenems, cotrimoxazole and piperacillin/tazobactam.” (Lines361 – 367)

“These combinations are most likely to cover gram-positive and -negative stains.” (Lines 372-373)

Further splitting the gram (+) and (-) group according to HAVO and CAVO sub-groups did not seem reasonable due to the resulting small group size. We hope you find the added data informative.

Round 2

Reviewer 1 Report

I recommend acceptance and publication of this manuscript.